# PROVABLE RL WITH EXOGENOUS DISTRACTORS VIA MULTISTEP INVERSE DYNAMICS

**Yonathan Efroni[1], Dipendra Misra[1], Akshay Krishnamurthy[1], Alekh Agarwal[2†], John Langford[1]**
[1]Microsoft Research, New York, NY
[2]Google

## ABSTRACT

Many real-world applications of reinforcement learning (RL) require the agent to deal with high-dimensional observations such as those generated from a megapixel camera. Prior work has addressed such problems with representation learning, through which the agent can provably extract endogenous, latent state information from raw observations and subsequently plan efficiently. However, such approaches can fail in the presence of temporally correlated noise in the observations, a phenomenon that is common in practice. We initiate the formal study of latent state discovery in the presence of such *exogenous* noise sources by proposing a new model, the Exogenous Block MDP (EX-BMDP), for rich observation RL. We start by establishing several negative results, by highlighting failure cases of prior representation learning based approaches. Then, we introduce the Predictive Path Elimination (PPE) algorithm, that learns a generalization of inverse dynamics and is provably sample and computationally efficient in EX-BMDPs when the endogenous state dynamics are near deterministic. The sample complexity of PPE depends polynomially on the size of the latent endogenous state space while not directly depending on the size of the observation space, nor the exogenous state space. We provide experiments on challenging exploration problems which show that our approach works empirically.

## 1 INTRODUCTION

In many real-world applications such as robotics there can be large disparities in the size of agent's observation space (for example, the image generated by agent's camera), and a much smaller latent state space (for example, the agent's location and orientation) governing the rewards and dynamics. This size disparity offers an opportunity: how can we construct reinforcement learning (RL) algorithms which can learn an optimal policy using samples that scale with the size of the latent state space rather than the size of the observation space? Several families of approaches have been proposed based on solving various ancillary prediction problems including autoencoding (Tang et al., 2017; Hafner et al., 2019), inverse modeling (Pathak et al., 2017; Burda et al., 2018), and contrastive learning (Laskin et al., 2020) based approaches. These works have generated some significant empirical successes, but are there provable (and hence more reliable) foundations for their success? More generally, what are the right principles for learning with latent state spaces?

In real-world applications, a key issue is robustness to noise in the observation space. When noise comes from the observation process itself, such as due to measurement error, several approaches have been recently developed to either explicitly identify (Du et al., 2019; Misra et al., 2020; Agarwal et al., 2020a) or implicitly leverage (Jiang et al., 2017) the presence of latent state structure for provably sample-efficient RL. However, in many real-world scenarios, the observations consist of many elements (e.g. weather, lighting conditions, etc.) with temporally correlated dynamics (see e.g. Figure 1 and the example below) that are entirely independent of the agent's actions and rewards. The temporal dynamics of these elements precludes us from treating them as uncorrelated noise, and as such, most previous approaches resort to modeling their dynamics. However, this is clearly wasteful as these elements have no bearing on the RL problem being solved.

---

†Work was done while the author was at Microsoft Research.
{yefroni, dimisra, akshaykr, jcl}@microsoft.com, alekhagarwal@google.com

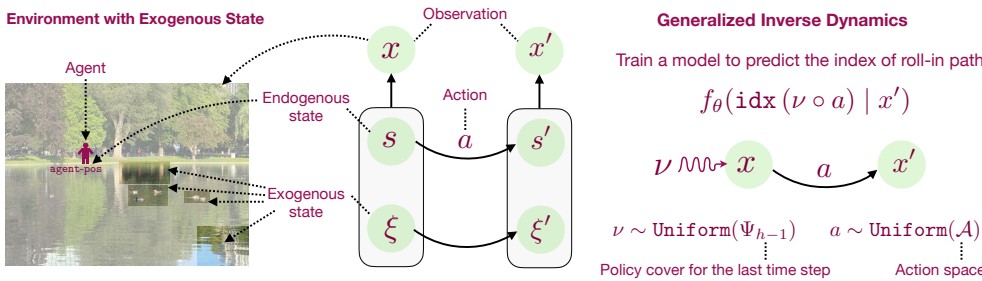

Figure 1: **Left:** An agent is walking next to a pond in a park and observes the world as an image. The world consists of a latent endogenous state, containing variable such as agent's position, and a much larger latent exogenous state containing variables such as motion of ducks, ripples in the water, etc. **Center:** Graphical model of the EX-BMDP. **Right:** PPE learns a generalized form of inverse dynamics that recovers the endogenous state.

As an example, consider the setting in Figure 1. An agent is walking in a park on a lonely sidewalk next to a pond. The agent's observation space is the image generated by its camera, the latent endogenous state is its position on the sidewalk, and the exogenous noise is provided by motion of ducks, swaying of trees and changes in lighting conditions, typically unaffected by the agent's actions. While there is a line of recent empirical work that aims to remove causally irrelevant aspects of the observation (Gelada et al., 2019; Zhang et al., 2020), theoretical treatment is quite limited (Dietterich et al., 2018) and no prior works address sample-efficient learning with provable guarantees. Given this, the key question here is:

*How can we learn using an amount of data scaling with just the size of the* endogenous *latent state, while ignoring the temporally correlated* exogenous *observation noise?*

We initiate a formal treatment of RL settings where the learner's observations are jointly generated by a latent endogenous state and an uncontrolled exogenous state, which is unaffected by the agent's actions and does not affect the agent's task. We study a subset of such problems called Exogenous Block MDPs (EX-BMDPs), where the endogenous state is discrete and decodable from the observations. We first highlight the challenges in solving EX-BMDPs by illustrating the failures of many prior representation learning approaches (Pathak et al., 2017; Misra et al., 2020; Jiang et al., 2017; Agarwal et al., 2020a; Zhang et al., 2020). These failure happen either due to creating too many latent states, such as one for each combination of ducks and passers-by in the example above leading to sample inefficiency in exploration, or due to lack of exhaustive exploration.

We identify one recent approach developed by Du et al. (2019) with favorable properties for EX-BMDPs with near-deterministic latent state dynamics. In Section 4 and Section 5, we develop a variation of their algorithm and analyze its performance. The algorithm, called Path Prediction and Elimination (PPE), learns a form of *multi-step inverse dynamics* by predicting the identity of the path that generates an observation. For near-deterministic EX-BMDPs, we prove that PPE successfully explores the environment using $O((SA)^2 H \log(|\mathcal{F}|/\delta))$ samples where $S$ is the size of the latent *endogenous* state space, $A$ is the number of actions, $H$ is the horizon and $\mathcal{F}$ is a function class employed to solve a maximum likelihood problem. Several prior works (Gregor et al., 2016; Paster et al., 2020) have also considered a multi-step inverse dynamics approach to learn a near optimal policy. Yet, these works do not consider the EX-BMDP model. Further, it is unknown whether these algorithms have provable guarantees, as PPE. Theoretical analysis of the performance of these algorithms in the presence of exogenous noise is an interesting future work direction.

Empirically, in Section 6, we demonstrate the performance of PPE and various prior baselines in a challenging exploration problem with exogenous noise. We show that baselines fail to decode the endogenous state as well as learning a good policy. We further, show that PPE is able to recover the latent endogenous model in a visually complex navigation problem, in accordance with the theory.

## 2   Exogenous Block MDP Setting

We introduce a novel *Exogenous Block Markov Decision Process* (EX-BMDP) setting to model systems with exogenous noise. We describe notations before formalizing the EX-BMDP model.

**Notations.** For a given set $\mathcal{U}$, we use $\Delta(\mathcal{U})$ to denote the set of all probability distributions over $\mathcal{U}$. For a given natural number $N \in \mathbb{N}$, we use the notation $[N]$ to denote the set $\{1, 2, \cdots, N\}$. Lastly, for a probability distribution $p \in \Delta(\mathcal{U})$, we define its support as $supp(p) = \{u \mid p(u) > 0, u \in \mathcal{U}\}$.

We start with describing the Block Markov Decision Process (BMDP) Du et al. (2019). This process consists of a finite set of observations $\mathcal{X}$, a set of *latent* states $\mathcal{Z}$ with cardinality $Z$, a finite set of actions $\mathcal{A}$ with cardinality $A$, a transition function $T : \mathcal{Z} \times \mathcal{A} \to \Delta(\mathcal{Z})$, an emission function $q : \mathcal{Z} \times \mathcal{A} \to \Delta(\mathcal{X})$, a reward function $R : \mathcal{X} \times \mathcal{A} \to [0, 1]$, a horizon $H \in \mathbb{N}$, and a start state distribution $\mu \in \Delta(\mathcal{Z})$. The agent interacts with the environment by repeatedly generating $H$-step trajectories $(z_1, x_1, a_1, r_1, \cdots, z_H, x_H, a_H, r_H)$ where $z_1 \sim \mu(\cdot)$ and for every $h \in [H]$ we have $x_h \sim q(\cdot \mid z_h)$, $r_h = R(x_h, a_h)$, and if $h < H$, then $z_{h+1} \sim T(\cdot \mid z_h, a_h)$. The agent does not observe the states $(z_1, \cdots, z_H)$, instead receiving only the observations $(x_1, \cdots, x_H)$ and rewards $(r_1, \cdots, r_H)$. We assume that the emission distributions of any two latent states are disjoint, usually referred as *the block assumption*: $supp(q(\cdot \mid z_1)) \cap supp(q(\cdot|z_2)) = \emptyset$ when $z_1 \neq z_2$. The agent chooses actions using a policy $\pi : \mathcal{X} \to \Delta(\mathcal{A})$. We also define the set of non-stationary policies $\Pi_{\text{NS}} = \Pi^H$ as a $H$-length tuple, with $(\pi_1, \cdots, \pi_H) \in \Pi_{\text{NS}}$ denoting that the action at time step $h$ is taken as $a_h \sim \pi_h(\cdot \mid x_h)$. The value $V(\pi)$ of a policy $\pi$ is the expected episodic sum of rewards $V(\pi) := \mathbb{E}_\pi[\sum_{h=1}^H R(x_h, a_h)]$. The optimal policy is given by $\pi^\star = \arg\max_{\pi \in \Pi_{\text{NS}}} V(\pi)$. We denote by $\mathbb{P}_h(x|\pi)$ the probability distribution over observations $x$ at time step $h$ when following a policy $\pi$. Lastly, we refer to an *open loop* policy as an element in all $\mathcal{A}^H$ sequences of actions. An open loop policy follows a pre-determined sequence of actions $\{a_1, .., a_H\}$ for $H$ time steps, unaffected by state information.

Given the aforementioned definitions, we define an EX-BMDP as follows:

**Definition 1** (Exogenous Block Markov Decision Processes)**.** *An EX-BMDP is a BMDP such that the latent state can be decoupled into two parts $z = (s, \xi)$ where $s \in \mathcal{S}$ is the endogenous state and $\xi \in \Xi$ is the exogenous state. For $z \in \mathcal{Z}$ the initial distribution and transition functions are decoupled, that is: $\mu(z) = \mu(s)\mu_\xi(\xi)$, and $T(z' \mid z, a) = T(s' \mid s, a)T_\xi(\xi' \mid \xi)$.*

The observation space $\mathcal{X}$ can be arbitrarily large to model which could be a high-dimensional real vector denoting an image, sound, or haptic data in an EX-BMDP. The endogenous state $s$ captures the information that can be manipulated by the agent. Figure 1, center, visualizes the transition dynamics factorization. We assume that the set of all endogenous states $\mathcal{S}$ is finite with cardinality $S$. The exogenous state $\xi$ captures all the other information that the agent cannot control and does not affect the information it can manipulate. Again, we make no assumptions on the exogenous dynamics nor on its cardinality $|\Xi|$ which may be arbitrarily large. We note that the block assumption of the EX-BMDP implies the existence of two inverse mappings: $\phi^\star : \mathcal{X} \to \mathcal{S}$ to map an observation to its endogenous state, and $\phi_\xi^\star : \mathcal{X} \to \Xi$ to map it to its exogenous state.

**Justification of assumptions.** The block assumption has been made by prior work (e.g., Du et al. (2019), Zhang et al. (2020)) to model many real-world settings where the observation is *rich*, i.e., it contains enough information to decode the latent state. The decoupled dynamics assumption made in the EX-BMDP setting is a natural way to characterize exogenous noise; the type of noise that is not affected by our actions nor affects the endogenous state but may have non-trivial dynamic. This decoupling captures the movement of ducks, captured in the visual field of the agent in Figure 1, and many additional exogenous processes (e.g., movement of clouds in a navigation task).

**Goal.** Our formal objective is reward-free learning. We wish to find a set of policies, we call a *policy cover*, that can be used to explore the entire state space. Given a policy cover, and for any reward function, we can find a near optimal policy by applying dynamic programming (e.g., Bagnell et al. (2004)), policy optimization (e.g., Kakade and Langford (2002); Agarwal et al. (2020b); Shani et al. (2020)) or value (e.g., Antos et al. (2008)) based methods.

**Definition 2** ($\alpha$-policy cover)**.** *Let $\Psi_h$ be a finite set of non-stationary policies. We say $\Psi_h$ is an $\alpha$-policy cover for the $h^{th}$ time step if for all $z \in \mathcal{Z}$ it holds that $\max_{\pi \in \Psi_h} \mathbb{P}_h(z|\pi) \geq \max_{\pi \in \Pi_{NS}} \mathbb{P}_h(z|\pi) - \alpha$. If $\alpha = 0$ we call $\Psi_h$ a policy cover.*

For standard BMDPs the policy cover is simply the set of policies that reaches each latent state of the BMDP (Du et al., 2019; Misra et al., 2020; Agarwal et al., 2020a). Thus, for a BMDP, the cardinality of the policy cover scales with $|\mathcal{Z}|$. The structure of EX-BMDPs allows to reduce the size of the

policy cover significantly to $|\mathcal{S}| \ll |\mathcal{Z}| = |\mathcal{S}| |\Xi|$ when the size of the exogenous state space is large. Specifically, we show that the set of policies that reach each *endogenous* state, and *do not depend on the exogenous* part of the state is also a policy cover (see Appendix B, Proposition 4).

## 3 FAILURES OF PRIOR APPROACHES

We now describe the limitation of prior RL approaches in the presence of exogenous noise. We provide an intuitive analysis over here, and defer a formal statement and proof to Appendix A.

**Limitation of Noise-Contrastive learning.** Noise-contrastive learning has been used in RL to learn a state abstraction by exploiting temporal information. Specifically, the HOMER algorithm (Misra et al., 2020) trains a model to distinguish between *real* and *imposter* transitions. This is done by collecting a dataset of quads $(x, a, x', y)$ where $y = 1$ means the transition was $(x, a, x')$ was observed and $y = 0$ means that $(x, a, x')$ was not observed. HOMER then trains a model $p_\theta(y \mid x, a, \phi_\theta(x'))$ with parameters $\theta$, on the dataset, by predicting whether a given pair of transition was observed or not. This provides a state abstraction $\phi_\theta : \mathcal{X} \to \mathbb{N}$ for exploring the environment. HOMER can provably solve Block MDPs. Unfortunately, in the presence of exogenous noise, HOMER distinguishes between two transitions that represent transition between the same latent endogenous states but different exogenous states. In our walk in the park example, even if the agent moves between same points in two transitions, the model maybe able to tell these transitions apart by looking at the position of ducks which may have different behaviour in the two transitions. This results in the HOMER creating $\mathcal{O}(|\mathcal{Z}|)$ many abstract states. We call this the *under-abstraction* problem.

**Limitation of Inverse Dynamics.** Another common approach in empirical works is based on modeling the inverse dynamics of the system, such as the ICM module of Pathak et al. (2017). In such approaches, we learn a representation by using consecutive observations to predict the action that was taken between them. Such a representation can ignore all information that is not relevant for action prediction, which includes all exogenous/uncontrollable information. However, it can also ignore controllable information. This may result in a failure to sufficiently explore the environment. In this sense, inverse dynamics approaches result in an *over-abstraction* problem where observations from different endogenous states can be mapped to the same abstract state. The over-abstraction problem was described at Misra et al. (2020), when the starting state is random. In Appendix A.3 we show inverse dynamics may over-abstract when the initial starting state is deterministic.

**Limitation of Bisimulation.** Zhang et al. (2020) proposed learning a bisimulation metric to learn a representation which is invariant to exogenous noise. Unfortunately, it is known that bisimulation metric cannot be learned in a sample-efficient manner (Modi et al. (2020), Proposition B.1). Intuitively, when the reward is same everywhere, then bisimulation merges all states into a single abstract state. This creates an *over-abstraction* problem in sparse reward settings, since the agent can falsely merge all states into a single abstract state until it receives a non-trivial reward.

**Bellman rank might depend on $|\Xi|$.** The Bellman rank was introduced in Jiang et al. (2017) as a complexity measure for the learnability of an RL problem with function approximations. To date, most of the learnable RL problems have a small Bellman rank. However, we show in Appendix A that Bellman rank for EX-BMDP can scale as $\mathcal{O}(|\Xi|)$. This shows that EX-BMDP is a highly non-trivial setting as we don't even have sample-efficient algorithms regardless of computationally-efficient.

In Appendix A we also describe the failures of FLAMBE (Agarwal et al., 2020a)) and autoencoding based approaches (Tang et al., 2017).

## 4 REINFORCEMENT LEARNING FOR EX-BMDPs

In this section, we present an algorithm *Predictive Path Elimination* (PPE) that we later show can provably solve any EX-BMDP with nearly deterministic dynamics and start state distribution of the endogenous state, while making no assumptions on the dynamics or start state distribution of the exogenous state (Algorithm 1). Before describing PPE, we highlight that PPE can be thought of as

---

**Algorithm 1** PPE($\delta, \eta$): Predictive Path Elimination

1: Set $\Psi_1 = \{\bot\}$, stochasticity level $\eta \leq \frac{1}{4SH}$           // $\bot$ denotes an empty path

2: **for** $h = 2, \ldots, H$ **do**

3:      Set $N = 16 \left( |\Psi_{h-1} \circ A| \right)^2 \log \left( \frac{|\mathcal{F}||\Psi_{h-1}|AH}{\delta} \right)$

4:      Collect a dataset $\mathcal{D}$ of $N$ *i.i.d.* tuples $(x, \upsilon)$ where $\upsilon \sim \text{Unf}(\Psi_{h-1} \circ \mathcal{A})$ and $x \sim \mathbb{P}(x_h \mid \upsilon)$.

5:      Solve multi-class classification problem: $\hat{f}_h = \arg\max_{f \in \mathcal{F}} \sum_{(x,\upsilon) \in \mathcal{D}} \ln f(\text{idx}(\upsilon) \mid x)$.

6:      **for** $1 \leq i < j \leq |\Psi_{h-1} \circ \mathcal{A}|$ **do**

7:          Calculate the path prediction gap: $\widehat{\Delta}(i,j) = \frac{1}{N} \sum_{(x,\upsilon) \in \mathcal{D}} \left| \hat{f}_h(i|x) - \hat{f}_h(j|x) \right|$.

8:          If $\widehat{\Delta}(i,j) \leq \frac{5/8}{|\Psi_{h-1} \circ \mathcal{A}|}$, then eliminate path $\upsilon$ with $\text{idx}(\upsilon) = j$. // $\upsilon_i$ and $\upsilon_j$ visit same state

9:      $\Psi_h$ is defined as the set of all paths in $\Psi_{h-1} \circ \mathcal{A}$ that have not been eliminated in line 8.

---

a computationally-efficient and simpler alternative to Algorithm 4 of Du et al. (2019) who studied rich-observation setting without exogenous noise.[1]

PPE performs iterations over the time steps $h \in \{2, \cdots, H\}$. In the $h^{th}$ iteration, it learns a policy cover $\Psi_h$ for time step $h$ containing open-loop policies. This is done by first augmenting the policy cover for previous time step by one step. Formally, we define $\Upsilon_h = \Psi_{h-1} \circ \mathcal{A} = \{\pi \circ a \mid \pi \in \Psi_{h-1}, a \in \mathcal{A}\}$ where $\pi \circ a$ is an open-loop policy that follows $\pi$ till time step $h - 1$ and then takes action $a$. Since we assume the transition dynamics to be near-deterministic, therefore, we know that there exists a policy cover for time step $h$ that is a subset of $\Upsilon_h$ and whose size is equal to the number of reachable states at time step $h$. Further, as the transitions are near-deterministic, we refer to an open-loop policy as a path, as we can view the policy as tracing a path in the latent transition model. PPE works by eliminating paths in $\Upsilon_h$ so that we are left with just a single path for each reachable state. This is done by collecting a dataset $\mathcal{D}$ of tuples $(x, \upsilon)$ where $\upsilon$ is a uniformly sampled from $\Upsilon_h$ and $x \sim \mathbb{P}_h(x \mid \upsilon)$ (line 4). We train a classifier $\hat{f}_h$ using $\mathcal{D}$ by predicting the index $\text{idx}(\upsilon)$ of the path $\upsilon$ from the observation $x$ (line 5). Index of paths in $\Upsilon_h$ are computed with respect to $\Upsilon_h$ and remain fixed throughout training. Intuitively, if $\hat{f}_h(i \mid x)$ is sufficiently large, then we can hope that the path $\upsilon_i$ visits the state $\phi^\star(x)$. Further, we can view this prediction problem as learning a multistep inverse dynamics model since the open-loop policy contains information about all previous actions and not just the last action. For every pair of paths in $\Upsilon_h$, we first compute a path prediction gap $\widehat{\Delta}$(line 7). If the gap is too small, we show it implies that these paths reach the same endogenous state, hence we can eliminate a single redundant path from this pair (line 8). Finally, $\Psi_h$ is defined as the set of all paths in $\Upsilon_h$ which were not eliminated. PPE reduces RL to performing $H$ standard classification problems. Further, the algorithm is very simple and in practice requires just a single hyperparameter ($N$). We believe these properties will make it well-suited for many problems.

**Recovering an endogenous state decoder.** We can recover a endogenous state decoder $\hat{\phi}_h$ for each time step $h \in \{2, 3, \cdots, H\}$ directly from $\hat{f}_h$ as shown below:

$$\hat{\phi}_h(x) = \min \left\{ i \mid \hat{f}_h(i \mid x) \geq \max_j \hat{f}_h(j \mid x) - \mathcal{O}(1/|\Upsilon_h|), i \in [|\Upsilon_h|] \right\}.$$

Intuitively, this assigns the observation to the path with smallest index that has the highest chance of visiting $x$, and therefore, $\phi^\star(x)$. We are implicitly using the decoder for exploring, since we rely on using $\hat{f}_h$ for making planning decisions. We will evaluate the accuracy of this decoder in Section 6.

**Recovering the latent transition dynamics.** PPE can also be used to recover a latent endogenous transition dynamics. The direct way is to use the learned decoder $\hat{\phi}_h$ along with episodes collected by PPE during the course of training and do count-based estimation. However, for most problems, recovering an approximate deterministic transition dynamics suffices, which can be directly read

---

[1]Alg. 4 has time complexity of $\mathcal{O}(S^4 A^4 H)$ compared to $\mathcal{O}(S^3 A^3 H)$ for PPE. Furthermore, Alg. 4 requires an upper bound on $S$, whereas PPE is adaptive to it. Lastly, Du et al. (2019) assumed deterministic setting while we provide a generalization to near-determinism.

from the path elimination data. We accomplish this by recovering a partition of paths in $\Psi_{h-1} \times \mathcal{A}$ where two paths in the same partition set are said to be *merged* with each other. In the beginning, each path is only merged with itself. When we eliminate a path $\upsilon_j$ on comparison with $\upsilon_i$ in line 8, then all paths currently merged with $\upsilon_j$ get merged with $\upsilon_i$. We then define an abstract state space $\widehat{\mathcal{S}}_h$ for time step $h$ that contains an abstract state $j$ for each path $\upsilon_j \in \Psi_h$. Further, we recover a latent deterministic transition dynamics for time step $h-1$ as $\hat{T}_{h-1} : \widehat{\mathcal{S}}_{h-1} \times \mathcal{A} \to \widehat{\mathcal{S}}_h$ where we set $\hat{T}_{h-1}(i,a) = j$ if the path $\upsilon_j \in \Psi_h$ gets merged with path $\upsilon_i' \circ a \in \Psi_h$ where $\upsilon_i' \in \Psi_{h-1}$.

**Learning a near optimal policy given a policy cover.** PPE runs in a reward-free setting. However, the recovered policy cover and dynamics can be directly used to optimize any given reward function with existing methods. If the reward function depends on the exogenous state then we can use the PSDP algorithm (Bagnell et al., 2004) to learn a near-optimal policy. PSDP is a model-free dynamic programming method that only requires policy cover as input (see Appendix D.1 for details). If the reward function only depends on the endogenous state, we can use a computationally cheaper value-iteration VI that uses the recovered transition dynamics. VI is a model-based algorithm that estimates the reward for each state and action, and performs dynamic programming on the model (see Appendix D.2 for details). In each case, the sample complexity of learning a near-optimal policy, given the output of PPE, scales with the size of endogenous and not the exogenous state space.

## 5 THEORETICAL ANALYSIS AND DISCUSSION

We provide the main sample complexity guarantee for PPE as well as additional intuition for why it works. We analyze the algorithm in near-deterministic MDPs defined as follows: Two transition functions $T_1$ and $T_2$ are $\eta$-*close* if for all $h \in [H], a \in \mathcal{A}, s \in \mathcal{S}_h$ it holds that $||T_1(\cdot \mid s,a) - T_2(\cdot \mid s,a)||_1 \leq \eta$. Analogously, two starting distribution $\mu_1$ and $\mu_2$ are $\eta$-close if $||\mu_1(\cdot) - \mu_2(\cdot)||_1 \leq \eta$. We emphasize that near-deterministic dynamics are common in real-world applications like robotics.

**Assumption 1** (Near deterministic endogenous dynamics). *We assume the endogenous dynamics is $\eta$-close to a deterministic model $(\mu_{D,\eta}, T_{D,\eta})$ where $\eta \leq 1/(4SH)$.*

We make a realizability assumption for the regression problem solved by PPE (line 5). We assume that $\mathcal{F}$ is expressive enough to represent the Bayes optimal classifier of the regression problems created by PPE.

**Assumption 2** (Realizability). *For any $h \in [H]$, and any set of paths $\Upsilon \subseteq \mathcal{A}^h$ with $|\Upsilon| \leq SA$ and where $\mathcal{A}^h$ denotes the set of all paths of length $h$, there exists $f_{\Upsilon,h}^\star \in \mathcal{F}$ such that: $f_{\Upsilon,h}^\star(\mathtt{idx}(\upsilon) \mid x) = \frac{\mathbb{P}_h(\phi^\star(x)) \mid \upsilon)}{\sum_{\upsilon' \in \Upsilon} \mathbb{P}_h(\phi^\star(x)) \mid \upsilon')}$, for all $\upsilon \in \Upsilon$ and $x \in \mathcal{X}$ with $\sum_{\upsilon' \in \Upsilon} \mathbb{P}_h(\phi^\star(x)) \mid \upsilon') > 0$.*

Realizability assumptions are common in theoretical analysis (e.g., Misra et al. (2020), Agarwal et al. (2020a)). In practice, we use expressive neural networks to solve the regression problem, so we expect the realizability assumption to hold. Note that there are at most $A^{S(H+1)}$ Bayes classifiers for different prediction problems. However, this is acceptable since our guarantees will scale as $\ln |\mathcal{F}|$ and, therefore, the function class $\mathcal{F}$ can be exponentially large to accommodate all of them.

We now state the formal sample complexity guarantees for PPE below.

**Theorem 1** (Sample Complexity). *Fix $\delta \in (0,1)$. Then, with probability greater than $1 - \delta$, PPE returns a policy cover $\{\Psi_h\}_{h=1}^H$ such that for any $h \in [H]$, $\Psi_h$ is a $\eta H$-policy cover for time step $h$ and $|\Psi_h| \leq S$, which gives the total number of episodes used by PPE as $\mathcal{O}\left(S^2 A^2 H \ln \frac{|\mathcal{F}|SAH}{\delta}\right)$.*

We defer the proof to Appendix C. Our sample complexity guarantees do not depend directly on the size of observation space or the exogenous space. Further, since our analysis only uses standard uniform convergence arguments, it extends straightforwardly to infinitely large function classes by replacing $\ln |\mathcal{F}|$ with other suitable complexity measures such as Rademacher complexity.

**Why does PPE work?** We provide an asymptotic analysis to explain why PPE works. Consider a deterministic setting and the $h^{th}$ iteration of PPE. Assume by induction that $\Psi_{h-1}$ is an exact policy cover for time step $h-1$. Therefore, $\Upsilon_h = \Psi_{h-1} \circ \mathcal{A}$ is also a policy cover for time step $h$. However, it may contain redundancies; it may contain several paths that reach the same endogenous state. We now show how a generalized inverse dynamics objective can eliminate such redundant paths.

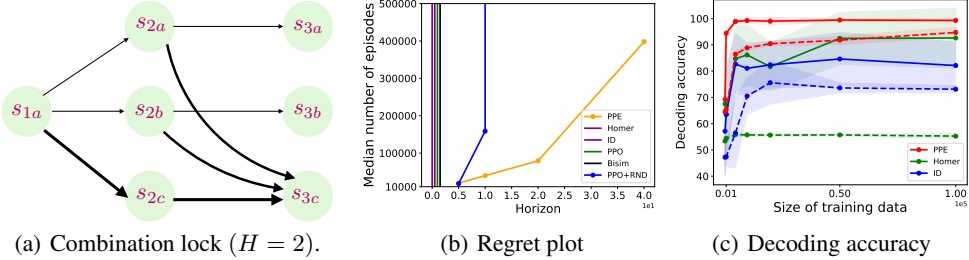

(a) Combination lock ($H = 2$).     (b) Regret plot     (c) Decoding accuracy

Figure 2: Results on combination lock. **Left:** We show the latent transition dynamics of combination lock. Observations are not shown for brevity. **Center:** Shows minimal number of episodes needed to achieve a mean regret of at most $V(\pi^\star)/2$. **Right:** State decoding accuracy (in percent) of decoders learned by different methods. Solid lines implies no exogenous dimension while dashed lines imply an exogenous dimension of 100.

Let $\mathbb{P}_h(\xi)$ denote the distribution over exogenous states at time step $h$ which is independent of agent's policy. The Bayes optimal classifier ($f_h^\star := f_{\Upsilon_h, h}$) of the prediction problem can be derived as:

$$f_h^\star(\texttt{idx}(\upsilon) \mid x) := \mathbb{P}_h(\upsilon \mid x) = \frac{\mathbb{P}_h(x \mid \upsilon)\mathbb{P}(\upsilon)}{\sum_{\upsilon' \in \Upsilon_h} \mathbb{P}_h(x \mid \upsilon')\mathbb{P}(\upsilon')} \overset{(a)}{=} \frac{\mathbb{P}_h(x \mid \upsilon)}{\sum_{\upsilon' \in \Upsilon_h} \mathbb{P}_h(x \mid \upsilon')} \overset{(b)}{=} \frac{\mathbb{P}_h(\phi^\star(x)) \mid \upsilon)}{\sum_{\upsilon' \in \Upsilon_h} \mathbb{P}_h(\phi^\star(x)) \mid \upsilon')},$$

where $(a)$ holds since all paths in $\Upsilon_h$ are chosen uniformly, and $(b)$ critically uses the fact that for any open-loop policy $\upsilon$ we have a factorization property,

$$\mathbb{P}_h(x \mid \upsilon) = q\left(x \mid \phi^\star(x), \phi_\xi^\star(x)\right) \mathbb{P}_h(\phi^\star(x) \mid \upsilon)\mathbb{P}_h(\phi_\xi^\star(x)).$$

Let $\upsilon_1, \upsilon_2 \in \Upsilon_h$ be two paths with indices $i$ and $j$ respectively. We define their exact path prediction gap as $\Delta(i, j) := \mathbb{E}_{x_h}\left[|f_h^\star(i \mid x_h) - f_h^\star(j \mid x_h)|\right]$. Assume that $\upsilon_1$ visits an endogenous state $s$ at time step $h$ and denote $\omega(s)$ as the number of paths in $\Upsilon_h$ that reaches $s$. Then $f_h^\star(i \mid x_h) = 1/\omega(s)$ if $\phi^\star(x_h) = s$, and 0 otherwise. If $\upsilon_2$ also visits $s$ at time step $h$, then $f_h^\star(i \mid x_h) = f_h^\star(j \mid x_h)$ for all $x_h$. This implies $\Delta(i, j) = 0$ and PPE will filter out the path with higher index since it detected both paths reach to the same endogenous state. Conversely, let $\upsilon_2$ visit a different state at time step $h$. If $x$ is an observation that maps to $s$, then $f_h^\star(i \mid x) = 1/\omega(s)$ and $f_h^\star(j \mid x) = 0$. This gives $|f_h^\star(i \mid x) - f_h^\star(j \mid x)| = 1/\omega(s) \geq 1/|\Upsilon_h|$ and, consequently, $\Delta(i, j) > 0$. In fact, we can show $\Delta(i, j) \geq \mathcal{O}(1/|\Upsilon_h|)$. Thus, PPE will not eliminate these paths upon comparison. Our complete analysis in the Appendix generalizes the above reasoning to finite sample setting where we can only approximate $f_h^\star$ and $\Delta$, as well as to EX-BMDPs with near-deterministic dynamics.

As evident, the analysis critically relies on the factorization property that holds for open-loop policies but not for arbitrary ones. This is the reason why we build a policy cover with open-loop policies.

## 6 EXPERIMENTS

We evaluate PPE on two domains: a challenging exploration problem called *combination lock* to test whether PPE can learn an optimal policy and an accurate state decoder, and a visual-grid world with complex visual representations to test whether PPE is able to recover the latent dynamics.

**Combination Lock Experiments.** The combination lock problem is defined for a given horizon $H$ by an endogenous state space $\mathcal{S} = \{s_{1,a}\} \cup \{s_{h,a}, s_{h,b}, s_{h,c}\}_{h=2}^H$, an exogenous state space $\Xi = \{0, 1\}^H$, an action space $\mathcal{A}$ with 10 actions, and a deterministic endogenous start state of $s_{1,a}$. For any state $s_{h,g}$ we call $g$ as its *type* which can be $a$, $b$ or $c$. States with type $a$ and $b$ are considered *good* states and those with type $c$ are considered *bad* states. Each instance of this problem is defined by two good action sequences $(a_h)_{h=2}^H, (a_h')_{h=2}^H$ with $a_h \neq a_h'$, which are chosen uniformly randomly and kept fixed throughout. At $h = 1$, the agent is in $s_{1,a}$ and action $a_1$ leads to $s_{2,a}$, $a_h'$ leads to $s_{2,b}$, and all other actions lead to $s_{2,c}$. For $h > 2$, taking action $a_h$ in $s_{h,a}$ leads to $s_{h+1,a}$ and taking action $a_h'$ in $s_{h,b}$ leads to $s_{h+1,b}$. In all other cases involving taking an action in a state $s_{h,g}$, we transition to the next bad state $s_{h+1,c}$. We visualize the latent endogenous dynamics in Figure 2a. The exogenous state evolves as follows. We set $\xi_1 \in \{0, 1\}^H$ where $\xi_1(i) \sim \texttt{Unf}(\{0, 1\})$ for each $i \in [H]$. At time step $h$, $\xi_h$ is generated from $\xi_{h-1}$ by uniformly flipping each bit in $\xi_{h-1}$ independently with

probability 0.1. There is a reward of 1.0 on taking the good action $a_{H,a}$ in $s_{H,a}$ and a reward of 0.1 on taking action $a_{H,b}$ in $s_{H,b}$. Otherwise, the agent gets a reward of 0. This gives a $V(\pi^\star) = 1$, and the probability that a random open loop policy gets this optimal return is $10^{-H}$.

An observation $x$ is generated stochastically from a latent state $z = (s, \xi)$. We map $s$ to a vector $w$ encoding the identity of the state. We concatenate $(w, \xi)$, add Gaussian noise to each dimension, and multiply the result by a Hadamard matrix to generate $x$. See Appendix F for full details. Our construction is inspired by prior work (Du et al., 2019; Misra et al., 2020).

**Baseline.** We compare PPE with five baselines on the combination lock problem. These include PPO (Schulman et al., 2017) which is an actor-critic algorithm, PPO + RND (Burda et al., 2019) which adds an exploration bonus to PPO using prediction errors, Homer that uses contrastive learning (Misra et al., 2020), and another algorithm ID which is similar to Homer but instead of contrastive learning it learns an inverse dynamics model to recover the state abstraction. Lastly, we also compare with Bisim that learns a bisimulation metric along with an actor-critic agent (Zhang et al. (2020)). We use existing publicly available codebases for these baselines. Our implementation of PPE very closely follows the pseudo-code in Algorithm 1. We model $\mathcal{F}$ using a two-layer feed-forward network with ReLU non-linearity. We train $\mathcal{F}$ with Adam optimization and use a validation set to do model selection. We refer readers to Appendix F for additional experimental details.

**Results.** Figure 2b shows results for values of $H$ in $\{5, 10, 20, 40\}$. For each value of $H$, we plot the minimal number of episodes $n$ needed to achieve a mean regret of at most $V(\pi^\star)/2 = 0.5$. We run each algorithm 5 times with different seeds and report the median performance. If an algorithm is unable to achieve the desired regret in $5 \times 10^5$ episodes we set $n = \infty$. We observe that PPO is unable to solve the problem at $H = 5$. PPO + RND is able to solve the problem at $H = 5$ and $H = 10$, showing the exploration bonus induced by random network distillation helps. However, it is unable to solve the problem for larger values of $H$. We observe that Homer and ID are also unable to solve the problem for any value of $H$. Bisim also fails to solve the problem for any $H \geq 5$. This agrees with the theoretical prediction that Bisim provides no learning signal when running in sparse-reward settings. In the absence of any reward, the bisimulation objective incentivizes mapping all observations to the same representation which is not helpful for further exploration. Lastly, PPE is able to solve the problem for all values of $H$ and is significantly more sample efficient than baselines. Since the reward function of the combination-lock problem depends only on the endogenous state, we run PPE and then a value-iteration like algorithm (see Appendix D.2) to learn a near optimal policy.

In order to understand the failure of Homer and ID, we investigate the accuracy of the state abstraction learned by these methods and compare that with PPE. We focus on the combination lock setting with $H = 2$ and evaluate the learned decoder for the last time step. As the state abstraction models are invariant to label permutation we use the following evaluation metric: given a learned abstraction for the endogenous state $\hat{\phi} : \mathcal{X} \to [N]$ we compute $1/m \sum_{i=1}^m \mathbf{1}\{\hat{\phi}(x_{i,1}) = \hat{\phi}(x_{i,2}) \Leftrightarrow \phi^\star(x_{i,1}) = \phi^\star(x_{i,2})\}$, where $\{x_{i,1}, x_{i,2}\}_{i=1}^m$ are drawn independently from a fixed distribution $D$ with good support over all states. We report the percentage accuracy in Figure 2c. When there is no exogenous noise, Homer is able to learn a good state decoder with enough samples while ID fails to learn, in accordance with the theory. On inspection, we found that ID suffers from the under-abstraction issue highlighted earlier as it has difficulty separating observations from $s_{3a}$ and $s_{3b}$. On adding exogenous noise, the accuracy of Homer plummets significantly. The accuracy of ID also drops but this drop is mild since unlike Homer, the ID objective is able to filter exogenous noise. Lastly, we observe that PPE is always able to learn a good decoder and is more sample efficient than baselines.

**Visual Grid World Experiments.** We test the ability of PPE to recover the latent endogenous transition dynamics in visual grid-world problem.[2] The agent navigates in a $N \times N$ grid world where each grid can contain a stationary object, the goal, or the agent. The agent's endogenous state is given by its position in the grid and its direction amongst four possible canonical directions. The agent can take five different actions for navigation. The world is visible to the agent as a $8N \times 8N$ sized RGB image. We add exogenous noise as follows: at the beginning of each episode, we independently sample position, size and color of 5 ellipses. The position and size of these ellipses is perturbed after each time step independent of the action. We project these ellipses on top of the world's image. Figure 3 shows sampled observations from the $7 \times 7$ gridworld that we experiment on. The

---

[2]We use the following popular gridworld codebase: https://github.com/maximecb/gym-minigrid

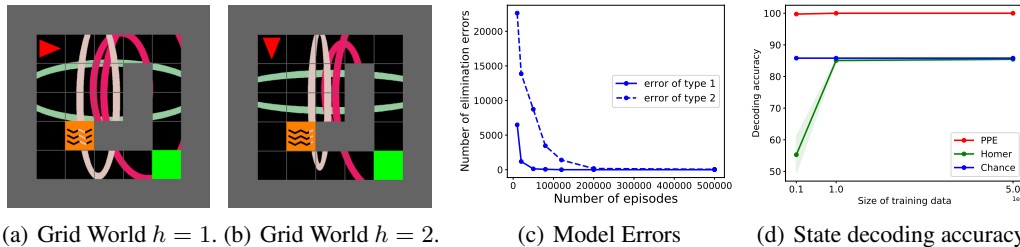

(a) Grid World $h = 1$. (b) Grid World $h = 2$.  (c) Model Errors  (d) State decoding accuracy

Figure 3: Results on visual grid world. **Left two:** Shows sampled observations for the first two steps from the visual gridworld domain. The agent is depicted as a red-triangle, lava in orange, walls in grey, and the goal in green. **Center Right:** Shows errors of type 1 and type 2 made by the PPE in recovering the latent endogenous dynamics. **Right:** State decoding accuracy of PPE, Homer and a random uniform decoder. (see Section 6)

exogenous state is given by the position, size and color of ellipses and is much larger than $|\mathcal{S}| \leq 4N^2$. We model $\mathcal{F}$ using a two-layer convolutional neural network and train it using Adam optimization. We defer the full details of setup to Appendix F.

Since the problem has deterministic dynamics, we evaluate the accuracy of the learned transition model by measuring it in terms of accuracy of the elimination step (Algorithm 1, line 8), since this step induces our algorithm's mapping from observations to endogenous latent states. For a fixed $h \in \{2, \cdots, H\}$, let $\nu_i$ and $\nu_j$ be two paths in $\Psi_{h-1} \circ \mathcal{A}$. We compute two type of errors. Type 1 error computes whether PPE merged these paths, i.e., predicted them as mapping to the same abstract state, when they go to *different endogenous* states. Type 2 error computes whether PPE predicted the paths as mapping to different abstract states, when they map to the *same endogenous* state. We report the total number of errors of both types by summing over all values of $h$ and all pairs of different paths in $\Psi_{h-1} \circ \mathcal{A}$. Type 1 errors are more harmful, since they can lead to exploration failure. Specifically, merging paths going to different states may result in the algorithm avoiding one of the two states when exploring at the next time step. Type 2 errors are less serious but lead to inefficiency due to using redundant paths for exploration.

**Results.** We report results on learning the model in in Figure 3c. We see that PPE is able to reduce the number of type 1 errors down to 0 using $2 \times 10^5$ episodes per time step. This is important since even a single type 1 error can cause exploration failures. Similarly, PPE is able to reduce type 2 errors and is able to get them down to 56 with $5 \times 10^5$ episodes. This is acceptable since type 2 errors do not cause exploration failures but only cause redundancy. Therefore, at $2 \times 10^5$ samples, the algorithm makes 0 type 1 errors and just a handful type 2 errors. This is remarkable considering that PPE compares roughly $2 \times 10^5$ pairs of paths in the entire run. Hence, it makes only $\leq 0.03\%$ type 2 errors. Further, the agent is able to plan using the learned transition model and receive the optimal return. We also evaluate the accuracy of state decoding on this problem. We compare the state decoding accuracy of PPE and Homer at $H = 2$ using an identical evaluation setup to the one we used for combination lock. Figure 3d shows the results. As expected, PPE rapidly learns a highly accurate decoder while Homer performs only as well as a random uniform decoder.

## 7 CONCLUSION

In this work, we introduce the EX-BMDP setting, an RL setting that models exogenous noise, ubiquitous in many real-world systems. We show that many existing RL algorithms fail in the presence of exogenous noise. We present PPE that learns a multi-step inverse dynamics to filter exogenous noise and successfully explores. We derive theoretical guarantees for PPE in near-deterministic setting and provide encouraging experimental evidence in support of our arguments. To our knowledge, this is the first such algorithm with guarantees for settings with exogenous noise. Our work also raises interesting future questions such as how to address the general setting with stochastic transitions, or handle more complex endogenous state representations. Another interesting line of future work direction is the analysis of other approaches that learn multi-step inverse dynamics (Gregor et al., 2016; Paster et al., 2020) and understanding whether these approaches can also provably solve EX-BMDPs.

ACKNOWLEGMENTS

We would like to thank the reviewers for their suggestions and comments. We acknowledge the help of Microsoft's GCR team for helping with the compute. YE is partially supported by the Viterbi scholarship, Technion.

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
