# OpenReview forum: "Provably Filtering Exogenous Distractors using Multistep Inverse Dynamics"
_ICLR.cc/2022/Conference — ICLR 2022 Oral_

### Official Review · Reviewer_fNfe · 2021-10-27

**Correctness:** 4
**Technical Novelty And Significance:** 3
**Empirical Novelty And Significance:** 3
**Recommendation:** 8
**Confidence:** 5

**Main Review:**

The paper is very well motivated. It has convinced me that being invariant to exogenous noise is important and that the EX-BMDP is useful abstraction for making this problem concrete. It is also very clearly presented. The argumentation is clear, and the flow of the paper is quite natural. Even the lengthy appendix is fairly easy to parse, with straightforward proofs and relevant details (e.g. specific claims about the unsuitability of alternative approaches).

One shortcoming is potential impact / applicability. I'm not convinced that environments with single starting states and near-deterministic dynamics is a very rich problem class. Or more specifically, I'm not convinced that restricting myself to this class would be preferable to working in the full space of MDPs with e.g. the possibility of aliasing a few states with a single-step inverse-dynamics approach. I am aware that impact is very hard to predict in advance, so I won't let this aspect unduly affect my scoring. And I'm convinced that even if PPE is never used in practice, this paper should still be accepted for its useful problem formulation and theoretical results. But a quick read would leave one with the impression that the alternatives (e.g. contrastive learning, inverse dynamics, bisimulation metrics) flaws outweigh their considerable advantages (e.g. applicability with stochastic dynamics) --  explicitly noting where alternative approaches are preferable would be appreciated.

I also noticed what I believe to be an omission in your related works: mutual information / empowerment based methods. "Variational Information Maximisation for Intrinsically Motivated Reinforcement Learning" seems particularly related in its search for predicable action sequences. And the follow-up work "Variational Intrinsic Control" appears to be a multi-step inverse dynamics analog that doesn't require open-loop policies. If you can show that these approaches are flawed in a way that PPE is not, then that would considerably raise my assessment of this work. And even if not, I believe this field (if not these specific works) should be acknowledged, as they similarly attempt to learn what in the environment is controllable.

While I agree that PPE is applicable in no-reward situations whereas bi-simulation metrics are not, your experiments all involve rewards, so I'm surprised a bi-simulation metric method was not included as a baseline.

While this isn't necessary for acceptance, it is worth noting that prior work has established much more challenging benchmarks for evaluating the representation of exogenous noise, and utilizing a pre-existing benchmark would make your empirical results considerably more impressive.

This is a small point, but it is not initially obvious why inverse dynamics fails on the combination lock problem. Does alliasing occur just when e.g. s_2a --> s_3a and s_2b-->s_3b have the same action?

**Summary Of The Paper:**

The authors propose a novel algorithm PPE, which they prove efficiently eliminates exogenous noise under certain assumptions (e.g. near deterministic dynamics). PPE works by growing a set of open-loop policies (action sequences) sufficient to reach all possible states for an increasing horizon. This is done by predicting the index of the policy from its final state, and eliminating states that are not sufficiently predictable. They show that both in theory and in practice that popular alternatives (noise contrastive and inverse-dynamics approaches) either fail to ignore exogenous noise or fail to distinguish between actually different states.

**Summary Of The Review:**

A well-argued paper that sets up an important problem and introduces a novel algorithm (PPE) to solve it. This is somewhat undercut by PPE only working in a restrictive setting. Empirical results would benefit from an additional baseline (bisimulation) and (optionally) a previously published benchmark. Some discussion of mutual information approaches should be made. But some reasonable attempt is made towards these improvements, I'd happily see this work accepted.

---

> ### Author Response · Authors · 2021-11-17
> **Comment of review of fNfe**
>
> We thank the reviewer for the detailed comments. We address the comments below.
>
> **1. The assumption on near deterministic endogenous dynamics.** We acknowledge the limitation of this assumption. However, we believe that the PPE algorithm is an important first step towards a general solution of the EX-BMDP model (and, more generally, to RL with exogenous noise).  The importance of the result stems from the fact PPE is the first provably efficient algorithm for such a setting. Furthermore, we believe that generalizing it to the fully stochastic setting is an important next step to take, and is an open problem posed by our work.
>
> $\quad$ We also note that contrastive learning and inverse dynamics approaches fail even in deterministic settings. Further, we believe that the failure of these approaches is quite common. E.g., the contrastive learning approaches get confused with the presence of any exogenous noise, whereas the aliasing of states for inverse dynamics as explained later in this response is quite common. Therefore, we believe that issues with the alternative approaches are quite serious and the fact that PPE can surmount them even in a deterministic setting provides an important lesson.
>
> $\quad$ Furthermore, we do want to stress the simplicity of the PPE algorithm. PPE is not only provably efficient but very simple to implement and run. It only requires access to a multi-class classification oracle, which is routinely used in practice. PPE is also adaptive to the number of latent states (it does not require an upper bound on the number of latent states, but calculates this quantity adaptively), and, lastly, PPE is very fast in practice. This makes PPE a strong algorithm to consider in cases where the near deterministic assumptions hold.
>
>
> **2. Reference to Variational Intrinsic Control.** We thank the reviewer for pointing these references. Indeed, _“Variational Intrinsic Control”_ is a form of multi-step inverse dynamics. We will cite and discuss it in the revision. However, the algorithm in that paper may fail in the presence of an exogenous distractor. One way to see a failure case is when the set $\Omega$ contains only exogenous states. That is, in the initialization step, the option-set may include only the exogenous states at time step t=2. If the sampling policy is uniform $\pi = U(A)$ (without prior knowledge that is the only reasonable choice), then it can be shown that: $I(\Omega, s_f | s_0) =  I( \Omega, \phi_{exo}^*(s_f) | \phi_{exo}^*(s_0))$ where $\phi_{exo}^*(s)$ maps a state to its exogenous state. This can be proved using the fact that $P_{a \sim U(A)}(s) = q(s| \phi_{endo}^*(s), \phi_{exo}^*(s))P_{a\sim U(A)}( \phi_{endo}^*(s)) P_{a\sim U(A)}( \phi_{exo}^*(s)),$ and the definition of mutual information.
>
> $\quad$ Thus, a fixed-point solution of algorithm 1 of _"Variational Intrinsic Control"_ is to set $\Omega$ as the set of exogenous states. This counterexample holds in the exact case, i.e., when we have access to exact computations. Nevertheless, it highlights the problematic nature of algorithm 1 of _"Variational Intrinsic Control"_ applied to the EX-BMDP model.
>
> **3. Comparison with Bisimulation.** Previous work has shown that bisimulation can fail even in the presence of reward. See [Modi et al. 2019](https://arxiv.org/pdf/1910.10597.pdf), Proposition B.1 and also see discussion of it in Section 3 in our paper. In particular, this happens in sparse-reward settings such as the combination lock experiments. Intuitively, the agent will not receive a reward for a long time in these settings. Hence, a degenerate solution that maps all observations to the same abstract state will receive optimal bisimulation loss. However, this degenerate solution results in poor exploration because of which the agent will not receive the reward. We will revise the paper to include empirical results on bisimulation on the combination lock.
>
> Reference: Sample Complexity of Reinforcement Learning using Linearly Combined Model Ensembles, Modi et al., 2019
>
> **4. State aliasing in inverse dynamics.** You are correct that inverse dynamics can merge state $s_{h, a}$ and $s_{h, b}$ for different values of h in the combination lock experiment. However, the reason for failure is more serious than accidental cases where a_{h-1} = a'_{h-1}. More generally, anytime two states have a disjoint set of _parent states_, then inverse dynamics can merge them. This happens since we can rely on the parent state to predict the action. As the parent states are disjoint, therefore, the parent information also uniquely identifies which of the two states is being addressed. E.g., even if we merge s_3a = s_3b to an abstract state $\bar{s}$, we can still predict $p(. $| s_2a, $\bar{s})$ and $p(. $| s\_2b, $\bar{s})$ correctly. This makes the failure of inverse dynamics quite common. We are happy to explain this in more detail if desired.

---

> > ### Author Response · Authors · 2021-11-21
> > **Thanks!**
> >
> > We have updated the paper to add results on bisimulation. We have also added the discussion of "Variational Intrinsic Control" Gregor et al., in related work. See text in blue in Section 3 and Section 6. As expected, bisimulation is unable to solve combination lock which has a sparse-reward structure. We have also added a conclusion section that discusses future work on the more general setting. If you have any further comments or questions, then we are happy to answer them by the Monday deadline for author discussion. However, if our comments resolve your doubts, then we will appreciate it if you can update your review.

---

> > ### Comment · Reviewer_fNfe · 2021-11-22
> > **Remaining VIC concerns**
> >
> > Thanks for the response!
> >
> > I'm generally happy with the explanations and changes to the paper.
> >
> > That said, I very much dispute your characterisation of variational intrinsic control (VIC). In VIC algo 1, the policy over actions is learned jointly with the rest of the objective, so your statement that the uniform random policy is "the only reasonable choice" doesn't make sense. Indeed, this policy will learn to ignore the exogenous states as they don't allow transmitting any information about which option is currently being executed. While the option space could certainly be made to have excessive entropy without harming the total mutual information, I would claim that the minimum entropy option distribution (for a set level of mutual information) would only contain the reachable endogenous states (and this can be approximated by known methods e.g. VALOR).
> >
> > That said, I wasn't really expecting you to refute this line of work in the timeframe of a rebuttal. I merely wanted it acknowledged as an alternative to be explored in future work. If the main text is changed to reflect this, then I'll raise my score to an 8 in light of your other substantial improvements.

---

> > > ### Author Response · Authors · 2021-11-22
> > > **Thanks for your feedback**
> > >
> > > Thank you for the response.
> > >
> > > We have revised the description of VIC to incorporate your comment (changes concerning your comment are in red color in Section 1 and Section 7). We have moved the discussion of these algorithms early into the introduction (see text in red color in Section 1). We understand that when an observation-dependent policy is used and jointly learned, our counter-example doesn't hold. We do think that analysis of VIC in the presence of exogenous noise is an interesting future work direction. We have also added a line explicitly in the conclusion section (see text in red color in Section 7) to consider VIC as a future work direction. Please let us know if you have any further comments, or suggestions for improvement.

---

> > > > ### Comment · Reviewer_fNfe · 2021-11-23
> > > > **Thanks for the rapid response!**
> > > >
> > > > Thanks for the change! Score updated 6-->8

---

### Official Review · Reviewer_yfQT · 2021-11-02

**Correctness:** 4
**Technical Novelty And Significance:** 3
**Empirical Novelty And Significance:** 3
**Recommendation:** 8
**Confidence:** 3

**Main Review:**

Strengths:

- The problem setting is important and seems to be relevant to many real-world problems.
- While the idea of using inverse dynamics and even multi-step inverse dynamics [1] to learn representations in these types of environments to learn controllable representations is not novel, this paper provides a theoretical basis for this choice, which I think is valuable.
- I found sections 1 through 5 to be very well written.

Weaknesses:

- In the experiment section, it seems as though you are testing the planning/policy learning abilities with PPE in both of the environments. However, as far as I can tell, there is no indication of how you perform this planning in PPE in the main text. The paper simply switches from talking about learning policy covers to showing regret for environments with only a small mention of the fact that you could plan using the data collected with PPE. The way the experiments are run and the choice to test the planning performance / how the state representations are obtained in experiments should be explained in more detail in my opinion.
- The paper discusses prior works with inverse dynamics seemingly only in the context of their applications in representation learning in two prior works on exploration. I think the application of multi-step inverse dynamics in goal-conditioned environments described in [1] is worth mentioning, since for a fixed goal this algorithm seems like it is doing something similar to PPL. Related is the lack of a "related works" section in the paper, which would be helpful for framing PPL in the context of prior works.
- There is no conclusion section, which I feel would improve the paper.

My major questions are:

1. How is the policy learning done in the experiment section? How are representations found in the decoding accuracy plot?
2. How does this paper relate to prior works on inverse dynamics?

[1] Paster, Keiran, et al. Planning from Pixels Using Inverse Dynamics Models. 2020. openreview.net, https://openreview.net/forum?id=V6BjBgku7Ro.

**Summary Of The Paper:**

This paper introduces a model called the Exogenous Block MDP (EX-BMDP) where the latent state contains both controllable (endogenous) and uncontrollable (exogenous) elements. The paper proposes an algorithm to find a policy cover with sample complexity that depends only on the size of the endogenous state rather than the observation or exogenous state. The algorithm works by training a classifier to predict the actions that were taken to get to a state. Since the exogenous state is not affected by actions, states for which the classifier returns similar results are likely the same endogenous state. The algorithm builds up a set of policies (sequences of actions are sufficient in the near-deterministic setting) which visit unique endogenous states by using the classifier to deduplicate redundant action sequences. Besides proving the sample efficiency of their approach, experiments are provided which show that this algorithm performs better than baseline approaches in terms of performance as well as in ability to decode the state from its representation in a simple combination lock environment as well as a grid world with distractors.

**Summary Of The Review:**

In my opinion, the contribution of the paper is strong since it analyzes an important problem setting, presents an analysis of representations learned using inverse dynamics, and shows that it works in practice. The weaknesses, specifically little discussion of prior work and problems with the experiment section seem fixable, and under the condition that these points improve, I will recommend acceptance.

---

> ### Author Response · Authors · 2021-11-17
> **Comment of review of yfQT**
>
> We thank the reviewer for the detailed comments. We address them below.
>
> **1. How is the policy learning done in the experiment section?** PPE returns a policy cover over the endogenous state space. PPE also learns an endogenous state decoder and estimates the latent endogenous transition dynamics. This gives us two existing dynamic programming-based approaches to do policy planning. When we have a general reward function, we can use PSDP to optimize the reward function (see Appendix D.1). PSDP can be run off the shelf using the policy cover returned by PPE. However, as PSDP is computationally expensive, we also discuss a value iteration based approach that can be employed when the reward only depends on the endogenous state (see Appendix D.2). For experiments in this paper (Figure 2 and 3), we use the value iteration algorithm. Due to the space limit, and since we felt that an application of value-iteration given a policy cover is quite standard, we chose to defer this part to the appendix. We will revise the paper to make this point clearer. We will also release the source code and along with detailed single pseudocode in the Appendix to provide the most clarity.
>
> **2. How are representations found in the decoding accuracy plot?** We discuss this in the paragraph called _“Recovering latent transition dynamics.”_ We extract a decoder for the endogenous state $\hat{\phi}_h$ for each time step $h$ from the learned classifier $\hat{f}_h$. This is the decoder we evaluate in Figure 2c.
>
> **3. How does this paper relate to prior works on inverse dynamics (Paster, Keiran, et al.)?** Thanks for the reference to Paster, Keiran, et al. We will cite and discuss this paper in the revision. We want to emphasize important differences between our work and theirs:
>
> - The algorithm of Paster, Keiran, et al., can recover goal states that depend on the exogenous state since they don't use any filtering mechanism to remove exogenous noise. Of course, they considered a different problem than the EX-BMDP setting, and therefore this issue was not problematic in their setting. When the goal states depend on such exogenous noise, the size of the latent state may significantly grow and scale with the cardinality of the exogenous state space instead of just the endogenous state space as for PPE.
>
> - The planning oracle used in their paper (see section 3.2 in their paper) is not computationally efficient, whereas ours is (the path elimination process we offer is computationally efficient and is polynomial in the number of latent states, actions and horizon).
>
> **4. Conclusion section.** We have added a conclusion in the revision.

---

> > ### Author Response · Authors · 2021-11-21
> > **Thanks!**
> >
> > We have updated the paper to add discussion of Paster, et al. We also now state how the state decoder is learned and how planning is done in greater detail. See text in blue in Section 3 and 4 and also in Appendix D.2.. Pseudocode with all details, as was run in our experiments, is stated in Appendix F.4. If you have any further comments or questions, then we are happy to answer them by the Monday deadline for author discussion. However, if our comments resolve your doubts, then we will appreciate it if you can update your review.

---

> > > ### Comment · Reviewer_yfQT · 2021-11-22
> > > **Thanks**
> > >
> > > Thank you for the response. The updates have addressed the majority of my concerns about the paper and I updated my review.

---

### Official Review · Reviewer_B1Cs · 2021-11-02

**Correctness:** 3
**Technical Novelty And Significance:** 2
**Empirical Novelty And Significance:** 3
**Recommendation:** 8
**Confidence:** 3

**Main Review:**

STRENGTHS

(S1) The problem of efficiently learning in high-dimensional observation spaces is a good one, and the authors provided an excellent discussion of the technical components of this problem in the early parts of the paper. I thought the definition and presentation of the EX-BMDP was particularly good.

(S2) I appreciate the authors providing experimental results to show that PPE performs well in some regards.

WEAKNESSES

(W1) I'm afraid I found the discussion of PPE to be a bit difficult to decode. Even looking at Algorithm 1, I find myself with a number of important questions unanswered, chief among them being "what about $\phi^*_e$"? It strikes me as odd that nowhere in this algorithm are the inverse mappings from observation to exogenous/endogenous state represented or used. Why is this?

(W2) I'm a little concerned with the computational tractability of the algorithm. My understanding from the paper is that one seeks some sort of "covering" of the state space--in what sense is it practical to obtain and/or store this covering? If the problem setting consists of high-dimensional observations this seems especially challenging.

(W3) From my naive perspective, it seems that the ultimate goal here is to use PPE and then actually perform reinforcement learning. However, I didn't understand from the brief discussion at the end of Section 4 (or, frankly, from reading the Appendix) how exactly that would be accomplished. I feel this is important enough that it should appear in the main paper.

POST-DISCUSSION COMMENTS

During the discussion, I feel as though the authors adequately addressed each of the issues I raised above, and so I'm happy to raise my score to accept.

**Summary Of The Paper:**

The authors consider the problem of reinforcement learning using high-dimensional observations (eg, images) that may contain both exogenous and endogenous state information. Seeking to remedy the issues with learning that arise due to the exogenous state information, the authors propose a new model called an Exogenous Block MDP (EX-BMDP) and a new algorithm called Predictive Path Elimination (PPE) to learn a generalization of the inverse dynamics of the EX-BMDP. Additionally, the authors present some experimental evidence that PPE performs well in the EX-BMDP setting compared to alternatives.

**Summary Of The Review:**

While I appreciate the importance of the problem setting and think I understand the general thrust of the paper here, I overall left the paper with a lot of confusion. I'm hoping the authors can provide clarity here--both in their response and by revising the manuscript--during the discussion phase. Indeed, after some discussion and edits to the paper, I feel that the paper is much more clear and recommend acceptance.

---

> ### Author Response · Authors · 2021-11-17
> **Comment on review of B1Cs**
>
> We thank the reviewer for the thorough comment. We will address the comments point-by-point.
>
> 1. **Inverse mapping $\phi^\star_e$:** We both use and learn this mapping. For example, we use it when expressing the Bayes optimal classifier in the main text (see the paragraph on *Why does PPE work?*), and we use it many times in the Appendix. One key feature of PPE is that it can learn $\phi^\star_e$. This is done implicitly in the Algorithm 1 pseudocode due to brevity. Basically, we can recover a learned decoder $\hat{\phi}_e$ from $\hat{f}$. We describe how to do this in the paragraph _"Recovering latent transition dynamics"_ (Section 4)  after the Algorithm 1 pseudocode. In this paragraph, the notation $\hat{\phi}_h$ represents the learned endogenous state decoder for time step $h$. We will change the notation to $\hat{\phi}^e_h$ to make this connection clearer. We also evaluate this decoder in Figure 2c. PPE is implicitly relying on this decoding since it is relying on $\hat{f}_h$ to make planning decisions. Learning this decoder has other uses such as debugging or visualizing the latent state.
>
>
> 2. **Tractability of PPE (important clarification).** We would like to clarify an important property and goal of our research. PPE is designed to achieve covering in the endogenous state space and so the sample and computational complexity are _bounded in terms of the size of endogenous state space_ which is finite, rather than the size of observation space or even the size of exogenous state space. Hence, PPE is ideally suited for very challenging high dimensional observation problems such as the visual grid world in Figure 3. In this case, the observation space is a 56x56x3 RGB image and each channel takes 255 values, hence the size of the observation space is bounded by $255^{56 \times 56 \times 3} \approx 10^{22000}$. In contrast, the size of endogenous state space is bounded by 25 x 4 = 100 (25 due to the position of agent in one of the 5x5 grid squares and 4 for each of the four directions). PPE’s sample and computational complexity are dependent on the size of endogenous state space (100) rather than the size of observation space ($10^{22000}$) or even the exogenous space. Further, note that PPE reduces RL to solving a sequence of $H$ classification tasks which are routinely performed in practice.
>
>
> 3. **Planning details**. We thank you for this comment. Once we have policy cover ($\Psi_h$) we can directly use existing methods such as PSDP to learn an optimal policy for any given reward function. We discuss how to do this in Appendix D.1 and provide pseudocode of PSDP. PSDP only relies on policy cover which is returned by PPE. In the special case, when the reward function is only dependent on the endogenous state, we can use a more efficient value iteration algorithm (see Appendix D.2). In light of this comment, we added a paragraph in section 4 (see _“learning a near optimal policy given a policy cover”_) and improved the writing in Appendix D.2. The value iteration algorithm is described in Algorithm 4. It tries to estimate the reward function $r(s, a)$ and, then returns the optimal policy with respect to this reward function and the recovered endogenous transition dynamics $T_D$. We will also release the source code and provide a detailed single pseudocode in the Appendix to provide more clarity. Please don’t hesitate to ask us for further clarification.

---

> > ### Comment · Reviewer_B1Cs · 2021-11-19
> > **Thanks**
> >
> > Thanks to the authors for responding to my comments. I believe (W1) and (W3) have been adequately addressed. For (W2), I greatly appreciate the clarification in the discussion here (i.e., the point about seeking a cover for the endogenous state space only)--it seems to me that the authors ought to make this more clear in the paper as well. In particular, it feels like something to this effect could easily be said somewhere at the bottom of p3.

---

> > > ### Author Response · Authors · 2021-11-21
> > > **Thanks for your response**
> > >
> > > Thank you for your response and suggestion!
> > >
> > > We have updated the paper and added a discussion for point W2 at the end of page 3, as you suggested. We emphasize that our goal (which PPE achieves) is to learn a policy cover on the endogenous state space with cardinality which is independent of both the observation space and exogenous space. If you have any further comments or questions, then we are happy to answer them by the Monday deadline for author discussion. If our comments resolve your doubts, then we will appreciate it if you can update your review.

---

### Official Review · Reviewer_Zamv · 2021-11-03

**Correctness:** 4
**Technical Novelty And Significance:** 3
**Empirical Novelty And Significance:** 2
**Recommendation:** 8
**Confidence:** 2

**Main Review:**

Strengths:

- EX-BMDPs seem to capture several popular simulated distractors that are currently being used in applied RL papers, such as random video backgrounds in MuJoCo tasks. I suspect the fact that the exogenous dynamics are relatively unconstrained will also make them a good model for practical tasks. The requirement for near-determinism in the endogenous dynamics is unfortunate, particularly given that it applies to the initial state distribution as well (often you want to perform well over a distribution of similar-but-not-identical tasks). However, I think this is not a major concern for exploration methods: so long as the possible initial states are reachable from some common starting state within $H$ steps, PPE will eventually explore them.
- The fact that it is possible to obtain a sample complexity bound completely independent of the complexity of $\Xi$ was surprising to me given the relatively complex form of exogenous noise in an EX-BMDP, and shows that the EX-BMDP class is at least tractable.
- The clarity of writing was appreciated, particularly when stating assumptions and comparing against past algorithms.

Weaknesses:

- I noticed that RND is missing from Section 3/Appendix A (shortcomings of existing methods), but does surprisingly well on the combination lock problem. Is it possible to give some intuition for where RND fails in general?
- The visual gridworld experiments are a little confusing. The takeaway seems to be that PPE drives accuracy to \~100% eventually on a less-toy problem. However, there are no baselines, so it is hard to determine how significant this achievement is. It would be useful to see how the baselines from Figure 2 fare in this setting, particularly in terms of sample efficiency.

**Summary Of The Paper:**

This paper proposes a new class of decision problems, the exogenous block-MDP (EX-BMDP), and an algorithm for acquiring minimal state representations for EX-BMDPs. EX-BMDPs are an extension of block MDPs that allows for additional "exogenous" state components that may have arbitrary Markovian dynamics, but cannot be influenced by actions, including through the actions' effects on "endogenous" state. The proposed algorithm, PPE, is able to recover a mapping from observed states to endogenous states (i.e. ignoring the latent distractor variables) in the setting where the endogenous state dynamics and initial state distribution are near-deterministic. It does this by comparing the set of policies that can reach each pair of states: informally, if $p(\pi \mid s_1) = p(\pi \mid s_2)$ for all policies $\pi$ (assuming a uniform $p(\pi)$), then $s_1$ and $s_2$ are treated as equivalent. In the deterministic case, policies reduce to action sequences ("paths"). Illustrative experiments on simple domains show that the proposed approach is able to quickly recover a state representation that depends mostly on the endogeneous component.

**Summary Of The Review:**

This paper proposes a general but tractable class of MDPs with exogenous distractors, and proposes a novel algorithm which provably obtains representations that exclude the spurious latent dimensions of the state. Given the technical significance of the work, I am in favour of acceptance.

-----

**Update:** the author response resolves the issue with visual gridworld experiments. I'm still in favour of acceptance.

---

> ### Author Response · Authors · 2021-11-17
> **Comment on review of Zamv**
>
> We thank the reviewer for the detailed and kind comment!
>
> -) We would like to emphasize that RND fails on the combination lock problem and underperforms PPE which not only can solve problem with longer horizon (H=40 for PPE vs H=10 for PPO+RND), but also that PPE uses fewer samples for H=10 than RND. Note that in Figure b, the number of episodes are on the y-axis and horizon is on the x-axis and the flatter the plot, the better it is. Regarding theoretical failures, it is difficult to provide a concise theoretical counterexample to RND. Our experiments though show that it is unlikely that PPO+RND enjoys the PAC-RL guarantees that we derive for PPE.
>
> -) While visual gridworld are more challenging in terms of representation, they are not as challenging as combination lock in terms of exploration. Therefore, our focus on this task is to recover the latent dynamics for which this task presents a non-trivial challenge due to its representation complexity. PPO and PPO+RND do not output latent model as a side-product, nor do they learn a state decoder. Hence, they are not valid baselines. We will revise the paper to add Homer and ID baselines on this task.

---

> > ### Author Response · Authors · 2021-11-21
> > **Thanks!**
> >
> > We have updated the paper to add baseline results on the visual grid world. If you have any further comments or questions, then we are happy to answer them by the Monday deadline for author discussion. If our comments resolve your doubts, then we will appreciate it if you can update your review.

---

### Author Response · Authors · 2021-11-17
**Comment for all reviewers**

Based on reviewer feedback, we have updated the paper to (new text is in blue colour for ease of reading):

1. Include extra experimental details. In particular, we report baseline results in the visual-grid world for state decoding (Figure 3d is new) as well as reporting a bisimulation-based baseline in Figure 2b.

2. Adding more details on how planning for a given reward is done, and how a transition model is extracted. These are in Section 4.

3. Related work discussion of prior work on multi-step inverse dynamics (Paster et al., and Gregory et al.). As discussed, these works cannot provably solve exogenous noise, unlike our algorithm PPE.

4. Adding a conclusion section that also discusses future work. Section 7.

---

### Decision · Program_Chairs · 2022-01-20

**Decision:**

Accept (Oral)

**Comment:**

The paper presents a new technique that infers the endogenous states of an RL problem, as well as the corresponding model and optimal policy.  A bound is derived that shows that the amount of data needed depends only on the number of endogenous states, while being independent of the number of exogenous states and the complexity of the observation space.  This is remarkable since this is the first technique that is shown to have a complexity that depends only on the number of endogenous states.  Furthermore, the bound derived is not just a theoretical bound.  It is a practical bound in the sense that it is used in the associated algorithm, which is demonstrated effectively on two problems.  Perhaps the main weakness of the paper is that no intuition is provided in the main paper to explain why the sample complexity can be made independent of the number of exogenous states and the complexity of the observation space.  The reader has to look at the proof in the supplementary material.  Nevertheless, this is remarkable work.